Effect of leaf position and days post-infiltration on transient expression of colorectal cancer vaccine candidate proteins GA733-Fc and GA733-FcK in Nicotiana benthamiana plant

Kim Kibum 1
http://orcid.org/0000-0003-2842-9530 Kang Yang Joo 1
http://orcid.org/0000-0002-1733-2453 Park Se Ra 1
Kim Do-Sun 2
Lee Seung-Won 1
Ko Kinarm 3
Ponndorf Daniel 4
Ko Kisung 1 ksko@cau.ac.kr
1 Department of Medicine, Therapeutic Protein Engineering Lab, College of Medicine, Chung-Ang University , Seoul , South Korea
2 Vegetable Research Division, National Institute of Horticultural and Herbal Science, Rural Development Administration , Wanju-gun , South Korea
3 Department of Stem Cell Biology, Konkuk University School of Medicine, Konkuk University , Seoul , South Korea
4 Department of Biological Chemistry, John Innes Centre, Norwich Research Park, Colney , Colney, Norwich , UK
Maddi Abhiram
Electronic publication date: 2021 Apr 7
Publication date: 2021
Volume: 9
Electronic Location ID: e10851
Received 2020 Aug 12; Accepted 2021 Jan 6
Copyright: © 2021 Kim et al.
Copyright year: 2021
Copyright holder: Kim et al.
License: This is an open access article distributed under the terms of the Creative Commons Attribution License, which permits unrestricted use, distribution, reproduction and adaptation in any medium and for any purpose provided that it is properly attributed. For attribution, the original author(s), title, publication source (PeerJ) and either DOI or URL of the article must be cited.
License URL: https://creativecommons.org/licenses/by/4.0/

Keywords: Agroinfiltration, Fusion protein, KDEL, Molecular biopharming, Recombinant vaccine

Funding: Korean Rural Development Administration Code# PJ0134372020 Korean Government (MEST) NRF-2017R1A2A2A0569788 Korean Government (MSIT) No.2019M3E5D5067214 This research was supported by a grant (Code# PJ0134372020) from the Korean Rural Development Administration, National Research Foundation of Korea Grant funded by the Korean Government (MEST) (NRF-2017R1A2A2A0569788), and the Bio & Medical Technology Development Program of the National Research Foundation (NRF) & funded by the Korean government (MSIT) (No.2019M3E5D5067214). The funders had no role in study design, data collection and analysis, decision to publish, or preparation of the manuscript.

==============================
Immunization with thetumor-associated antigen GA733 glycoprotein, which is highly expressed in colorectal cancer, is considered to be a promising strategy for cancer prevention and treatment. We cloned a fusion gene of GA733 and immunoglobulin Fc fragment (GA733-Fc), and that of GA733-Fc and an endoplasmic reticulum retention motif (GA733-FcK) into the Cowpea mosaic virus (CPMV)-based transient plant expression vector, pEAQ-HT. Agrobacterium tumefaciens (LBA4404) transformed with the vectors pEAQ-HT-GA733-Fc and pEAQ-HT-GA733-FcK was infiltrated into the leaves of Nicotiana benthamiana plants. To optimize harvesting of leaf to express therapeutic glycoproteins both spatially and temporally, protein expression levels at various leaf positions (top, middle, and base) and days post-infiltration (dpi) were investigated. The GA733-Fc and GA733-FcK genes were detected in leaves at 1–10 dpi using PCR. As assessed by western blot, GA733-Fc and GA733-FcK were expressed at the highest levels in the top leaf position at 5 dpi, and GA733-FcK was expressed more than GA733-Fc. The proteins were successfully purified from infiltrated N. benthamiana leaves using protein A affinity chromatography. ELISA verified that an anti-GA733 antibody recognized both purified proteins. Thus, a functional GA733-Fc colorectal cancer vaccine protein can be transiently expressed using a CPMV virus-based vector, with an optimized expression time and leaf position post-infiltration.

Introduction

Tumor-associated antigens (TAAs) are promising immunotherapeutic vaccine candidate proteins for cancer prevention and treatment (Lu et al., 2012; Wong-Arce, González-Ortega & Rosales-Mendoza, 2017). Immune responses for induction of immunoglobulins to target TAAs have been actively studied (Son, In & Pyo, 2005). Among TAAs, glycoprotein GA733, or epithelial cell adhesion molecule (EpCAM), is highly expressed on the surface of colorectal cancer cells and has been extensively investigated (Eyvazi et al., 2018; Lu et al., 2012). Anti-GA733 antibody has been shown to inhibit the growth of colorectal cancer cells, which results from GA733 immunization in mice (Brodzik et al., 2008). The extracellular domain of GA733 is often used as a target for colorectal cancer vaccination and is fused with a human immunoglobulin Fc fragment (GA733-Fc) to enhance immune responses (Kim, Song & Ko, 2018; Lee & Ko, 2017). The presence of the Fc domain markedly increases the plasma half-life of the protein, which prolongs therapeutic activity (Czajkowsky et al., 2012), and it simplifies purification of the protein by binding to protein A (Lee et al., 2020a; Park et al., 2020).

Plants have several advantages over existing mammalian cell-based systems in terms of lower production costs and the absence of human pathogen contamination (Chen et al., 2013; Lee & Ko, 2017; Loh & Wayah, 2014; Shin et al., 2019). In addition, plants require a simple cultivation environment (sunlight, water, nutrients, and fertilizer) to grow (Lim et al., 2015; Rigano & Walmsley, 2005). Moreover, unlike bacterial cells, plants have post-translational modification systems similar to those of mammalian cells, such as glycosylation. Since plant cells are eukaryotic and have an endomembrane system and secretory pathway (Loh & Wayah, 2014), plants can produce and assemble correctly folded and complex mammalian proteins in their cells (Menassa, Ahmad & Joensuu, 2012; Kim et al., 2020 (IJMS); Lee et al., 2020b (Plants)). Thus, plants are considered as an attractive alternative for the production of therapeutic recombinant proteins (Lee & Ko, 2017; Nausch et al., 2012). In our previous study, plants have been determined to highly produce recombinant GA733-Fc fusion proteins (Park et al., 2015).

In general, there are two approaches to express recombinant proteins in plants, which are transgenic and transient plant expression (Chen et al., 2013; Joh, McDonald & VanderGheynst, 2006; Ma, Li & Wang, 2019; Wydro, Kozubek & Lehmann, 2006). For transient expression, the vector pEAQ-HT, which is based on the Cowpea mosaic virus (CPMV), has been reported to be an efficient expression vector for therapeutic recombinant proteins (Shah, Almaghrabi & Bohlmann, 2013). This deconstructed vector lacks the viral coat protein and has been shown to allow high and fast recombinant protein production (Chen et al., 2013; Daniell et al., 2009; Peyret & Lomonossoff, 2013). In addition, Agrobacterium-mediated transient expression using a syringe or vacuum machine to transfer the CPMV-based pEAQ vector can be faster, cheaper, and more convenient than traditional stable transformation systems, without the need for viral replication (Loh & Wayah, 2014). However, the expression period and plant tissue position can influence both the quality and quantity of the recombinant protein (Lim et al., 2015; Yamamoto et al., 2018).

Thus, the goal of our current research is to establish spatial and temporal conditions for the maximum production of the colorectal cancer vaccine candidate proteins GA733-Fc and GA733-Fc with an ER retention motif (GA733-FcK), using the CPMV-based pEAQ transient expression system. In order to study optimal expression of both GA733-Fc and GA733-FcK, the GA733-Fc and GA733-FcK expression vectors were transferred to plant leaves in a laboratory scale using a direct-syringe agroinfiltration method. Gene insertion, protein expression, protein purification, and antibody recognition of both recombinant proteins were investigated along with day post-infiltration and leaf position.

Materials and Methods

Plasmid construction

To construct the pEAQ vectors for transient expression of GA733, the sequence encoding the GA733 was ligated to that of human IgG Fc to generate the GA733-Fc fusion gene, and this recombinant gene was tagged with an ER retention signal (KDEL) to generate GA733-FcK (Fig. 1). These sequences were cloned individually into the pEAQ-HT vector (Peyret & Lomonossoff, 2013; Sainsbury, Thuenemann & Lomonossoff, 2009), which is based on CPMV, a non-enveloped plant virus in the Comovirus family consisting of RNA-1 (6.0 kb) and RNA-2 (3.5 kb) molecules (Peyret & Lomonossoff, 2013). The T-DNA region of the pEAQ-HT vector possesses the suppressor of gene silencing, P19, and the gene encoding neomycin phosphotransferase III for resistance to kanamycin (Sainsbury, Thuenemann & Lomonossoff, 2009). The GA733-Fc and GA733-FcK sequences were cloned into the AgeI site of pEAQ-HT using the In-Fusion® HD Cloning Kit (Takara, Mountain View, CA, USA) (Fig. 1). In the recombinant pEAQ-HT vectors, expression of GA733-Fc and GA733-FcK was under the control of the Cauliflower mosaic virus 35S promoter and nopaline synthase terminator (Sainsbury, Thuenemann & Lomonossoff, 2009) (Fig. 1). Agrobacterium tumefaciens (LBA4404) was transformed with the recombinant binary plasmid vectors, and the resulting transformants were named Ag/pEAQ-GA733-Fc and Ag/pEAQ-GA733-FcK.

Figure 1 Schematic diagram of transient expression vectors for GA733-Fc (A) and GA733-FcK (B) proteins.

GA733-Fc and GA733-FcK gene expression cassettes were introduced into the pEAQ-HT plant transient expression vector. 35SP, Cauliflower mosaic virus 35S promoter; 5UTR, Cowpea mosaic virus RNA-2 5′-UTR; 3UTR, Cowpea mosaic virus RNA-2 3′-UTR; P, suppressor of gene silencing (P19); NII, kanamycin resistance gene (NPTII); K, endoplasmic reticulum retention signal (KDEL); NOST, nopaline synthase gene terminator; 35ST, Cauliflower mosaic virus 35S terminator. Expected protein structure of the recombinant fusion proteins GA733-Fc and GA733-FcK: green oval-shaped bar, GA733; connected yellow bar, Fc domain; blue star-shaped region, KDEL. Expected glycan structure: the symbols of the glycan structures are as follows: N-acetylglucosamine, gray square; mannose, white circle; overlapped white diamonds, α(1,3)-fucose; white triangle, β(1,2)-xylose.

Plants for agroinfiltration

Nicotiana benthamiana plant was used for agroinfiltration. The seeds of N. benthamiana were sown in commercial, nitrogen-fertilized soil (Seoul Bio, Chungcheongbuk-do, Korea). After 14 days, the seedlings were transplanted into pots containing nitrogen-fertilized soil. They were grown in a plant chamber at 25°C with 16 h of light and 8 h of darkness. Water was supplied every 2 days (50–100 ml). The plants were used for experiments after 6 weeks.

Agroinfiltration using the syringe method

Agrobacteria carrying each vector pEAQ-GA733-Fc and pEAQ-GA733-FcK were subcultured on LB agar containing antibiotics (10 μg/ml kanamycin) at 28 °C for 48 h. Then, 100 μl of infiltrating medium was inoculated in 40 ml of fresh LB medium with 10 μg/ml kanamycin and incubated for 21–23 h at 28 °C with shaking at 210 rpm until 0.35 optical density at 600 nm was obtained. The agrobacteria were pelleted by centrifugation at 3,000 rpm followed by resuspension in 40 ml of MMA buffer (10 mM 2-N-morpholinoethanesulfonic acid, pH 5.6, 10 mM MgCl2, 100 μM acetosyringone) (Sainsbury, Thuenemann & Lomonossoff, 2009). The abaxial side of each N. benthamiana leaf was pricked using a syringe needle to aid infiltration. Then, MMA buffer containing Ag/pEAQ-GA733-Fc or Ag/pEAQ-GA733-FcK was infiltrated into the undersides of each leaf with soft pressure using a 1 ml syringe (needle removed) until the entire leaf was wet (Loh & Wayah, 2014). The plant leaves with the same age were used for agrobacterium inoculation. The same leaf size was used for each location grown during the same period. After infiltration, the plants were grown at 25 °C in a chamber and watered every 1–2 days to prevent drying. The infiltrated leaves were harvested as described directly below. All samples were taken in triplicate with the same leaf at the same location for each dpi.

Harvesting agroinfiltrated leaves in accordance with days post-infiltration (dpi)

Nicotiana benthamiana leaf samples were collected at 1, 3, 5, 7, 8, 9, and 10 dpi to determine time-dependent recombinant protein expression levels. The infiltrated top, middle, and base leaves were harvested to investigate spatial protein expression and morphological changes.

Protein extraction and western blot analysis

The agroinfected leaves were harvested according to leaf positions (top, middle, and base leaves) for western blot analysis. After harvesting, the leaf blades were washed with tap water and gently dried with tissue paper. The harvested leaves (100 mg) were homogenized in three volumes (w/v) of 1× PBS (2.7 mM KCl, 137 mM NaCl, 10 mM Na2HPO4 and 2 mM KH2PO4) to extract the TSP. Two microliters of each sample was mixed with 0.4 μl of loading buffer (1 M Tris-HCl, 10% SDS, 5% 2-mercaptoethanol, 50% glycerol, and 0.1% bromophenol blue) and 16.6 μl of distilled water. The mixed samples were subjected to 10% SDS-PAGE in SDS running buffer (0.2 M glycine, 25 mM Tris-HCl, 0.1% [w/v] SDS). The protein gel was transferred to a nitrocellulose membrane (Millipore Corp, Billerica, MA, USA). The blotted membranes were transferred to 1× PBS-T buffer (1× PBS with 0.5% [v/v] Tween 20) and then blocked in 5% skim milk (Sigma, St. Louis, MO, USA) for 2 h at room temperature. The membrane was incubated with primary mouse anti-EpCAM mAb (anti-GA733 mAb) (R&D Systems, Minneapolis, MN, USA) (Isotype IgG2a, 1:5,000 dilution) to detect the GA733 portion of GA733-Fc and GA733-FcK, and then with 1:5,000-diluted secondary HRP-conjugated anti-mouse IgG2a (Abcam, Cambridge, UK), each for 2 h at room temperature with shaking (110 rpm). The protein bands were detected by using SuperSignal Chemiluminescent Substrate (Pierce, Rockford, IL, USA) and visualized by exposure to X-ray film (Fuji, Tokyo, Japan). EpCAM-Fc from mammalian cell (GA733-FcM) (rhEpCAM/Fc Chimera; R&D Systems, Minneapolis, MN, USA) was used as a positive control, and non-transformed N. benthamiana leaves were used as a negative control (Lim et al., 2015).

Genomic DNA isolation and PCR

Genomic DNA in transfected leaves was isolated using the Genomic DNA Mini Kit (Favorgen, Taiwan), according to the manufacturer’s instructions. PCR was conducted to confirm the presence of the GA733-Fc and GA733-FcK genes in the agroinfected leaves using a PCR premix kit (iNtRON Biotechnology, Seoul, Korea). The sequences of the primers were as follows: GA733-Fc forward primer, 5′-CAA ATT CGC GAC CGG ATG GCT ACT CAA CGA AGG-3′ and reverse primer, 5′-GGT GAT GCA TAC CGG TCA ACC CGG GGA CAG GGA-3′; GA733-FcK forward primer, 5′-CAA ATT CGC GAC CGG ATG GCT ACT CAA CGA AGG-3′ and reverse primer, 5′-GGT GAT GCA TAC CGG TCA GAG TTC ATC TTT ACC C-3′. The PCR consisted of an initial denaturation at 94 °C for 20 min, followed by 30 cycles of denaturation at 94 °C for 20 s, annealing at 64 °C for 30 s, and extension at 72 °C for 2 min, and then a final extension at 72 °C for 2 min. Non-transfected plants were used as a negative control, while vectors carrying the expression cassettes of GA733-Fc and GA733-FcK in Escherichia coli were used as positive controls.

Purification of GA733-IgG Fc and GA733-IgG FcK proteins from agroinfected leaves

To purify GA733-Fc and GA733-FcK, 100 g of agroinfected leaves were homogenized in extraction buffer (37.5 mM Tris-HCl, pH 7.5, 50 mM NaCl, 15 mM EDTA pH 8.0, 75 mM sodium citrate, and 0.2% sodium thiosulfate) using an aluminum blender (Hanil, Seoul, Korea) and then centrifuged at 7,000×g for 30 min at 4 °C. The supernatant was collected, further clarified using Miracloth (Biosciences, La Jolla, CA, USA), and adjusted to pH 5.1 with acetic acid. The protein solution was then centrifuged at 10,000×g for 30 min. The supernatant was collected, further clarified using Miracloth, and adjusted to pH 7.0 with 3 M Tris-HCl. Eight percent of the total supernatant volume of ammonium sulfate was added to the supernatant, and the solution was incubated at 4 °C for 2 h, and then 22.6% ammonium sulfate was added to the total solution volume. The mixture was incubated overnight at 4 °C and then centrifuged at 8,000×g for 30 min at 4 °C. The pellet was resuspended in 1/10 of the original volume of extraction buffer, and then centrifuged again at 8000×g for 30 min at 4 °C. The supernatant was filtered through a 0.45-μm syringe filter (MILLEX, Darmstadt, Germany) and loaded on a protein A resin column (Sepharose 4 Fast Flow, GE Healthcare, Sweden, NJ, USA). The target proteins were eluted using elution buffer according to the manufacturer’s recommendations. The final eluted protein fractions were subjected to SDS-PAGE to assess their purity. The gel was stained with Coomassie blue staining solution [10% acetic acid (v/v), 30% methanol (v/v), and 0.01% Coomassie blue (w/v)] at room temperature for 30 min with shaking. The gel was destained with destaining solution (10% acetic acid and 30% methanol) thrice for 30 min each with shaking. The stock GA733-Fc and GA733-FcK protein solutions were dialyzed three times against sterile 1× PBS.

ELISA to confirm the ability of GA733-Fc and GA733-FcK to bind anti-GA733 antibody

MaxiSorp™ 96-well ELISA plates (Nunc, Rochester, NY, USA) were coated with 100 μl per well of carbonate-bicarbonate (Sigma-Aldrich, St. Louis, MO, USA) buffer containing mouse anti-EpCAM mAb (anti-GA733 mAb) (R&D Systems, Minneapolis, MN, USA) (50 ng) and were incubated overnight at 4 °C. The plates were washed four times with 1× PBS-T [1× PBS plus 0.5% (v/v) Tween 20]. After washing, serially diluted GA733-Fc and GA733-FcK protein samples (12.5–0.196 ng) were added to each well, and then the plates were incubated for 2 h at 37 °C. The wells were washed four times with 1× PBS-T. One hundred microliters of HRP-conjugated goat anti-human IgG Fc antibody (Jackson, West Grove, PA, USA; diluted 1:5,000) was added to each well as a secondary antibody. After incubating for 2 h at room temperature, 3,3′,5,5′-tetramethylbenzidine (TMB) substrate solution (Seracare, Milford, MA, USA) was added to each well, and the color was allowed to develop for 3 min. TMB Stop Solution (Seracare, Milford, MA, USA) was used to stop the reaction. The absorbance of each well was read at 450 nm on a Gen5 microplate reader with version 2.01 software (Biotek, Winooski, VT, USA).

Results

Morphological changes in top, middle, and base leaves after agroinfiltration

To investigate morphological changes in the plant leaves after agroinfiltration, the leaf morphology of N. benthamiana plants agroinfiltrated with Agrobacterium carrying pEAQ-GA733-Fc (Ag/pEAQ-GA733-Fc) and Ag/pEAQ-GA733-FcK was examined according to days post-infiltration (dpi) and leaf position (top, middle, and base) (Fig. 2). In plants agroinfiltrated with Ag/pEAQ-GA733-Fc, leaf chlorosis was not observed (top, middle, and base leaves) until 3 dpi (Fig. 2A). The base leaves started to show chlorosis at 5 dpi. The middle and top leaves showed chlorosis at 7 dpi (Fig. 2A). Serious necrosis was observed on the base leaves at 8 dpi, while necrosis was observed on the middle and top leaves at 9 and 10 dpi, respectively. In plants agroinfiltrated with Ag/pEAQ-GA733-Fc and Ag/pEAQ-GA733-FcK, chlorosis and necrosis began on the base leaves at 5 and 7 dpi, respectively (Fig. 2B). The middle and top leaves both showed chlorosis and necrosis at 7 and 8 dpi, which was 2–3 days later than the base leaves (Fig. 2B). At 10 dpi, all the base leaves agroinfiltrated with both vectors showed necrosis (Fig. 2). After 7 dpi, regardless of the transgene, all the base leaves showed chlorosis and necrosis with wilting, and the leaf mass decreased rapidly (Fig. 2).

Figure 2 Physical appearance of N. benthamiana leaves (top, middle, and base leaves) infiltrated with Agrobacterial inoculum carrying pEAQ-GA733-Fc (Ag/pEAQ-GA733-Fc) (A) and pEAQ-GA733-FcK (Ag/pEAQ-GA733-FcK) (B) for 1, 3, 5, 7, 8, 9, and 10 days post-infec.

N. benthamiana plants were grown for 4–5 weeks in a greenhouse at 24 °C and 30% humidity, with 16 h light and 8 h dark. Six-week-old plants with similar number of leaves were used in this study.

PCR confirmation of GA733-Fc and GA733-FcK genes in agroinfiltrated leaves of N. benthamiana

PCR analysis was conducted to confirm the presence of the genes encoding GA733-Fc and GA733-FcK in the agroinfiltrated leaves at each leaf position (top, middle, and base leaves) (Fig. 3). The expected sizes of the amplified GA733-Fc (1,461 bp) and GA733-FcK (1,473 bp) bands were observed regardless of leaf position (Fig. 3). There was no band in the genomic DNA sample from a plant not infiltrated with Agrobacterium (Fig. 3). In plants agroinfiltrated with Ag/pEAQ-GA733-Fc, the top and middle leaves produced PCR bands at all observed dpi (Fig. 3A). All the base leaves at 1, 3, 5, and 7 dpi produced the PCR band. In plants agroinfiltrated with Ag/pEAQ-GA733-FcK, the PCR band trends were similar to those with Ag/pEAQ-GA733-Fc (Fig. 3B). The agroinfiltrated base leaves at 8, 9, and 10 dpi were not included for PCR analysis owing to the lack of sample mass caused by severe chlorosis, necrosis, and wilting.

Figure 3 PCR analysis of genomic DNA from plant leaves (top, middle, and base) at days post infiltration (dpi) with Ag/pEAQ-GA733-Fc and Ag/pEAQ-GA733-FcK.

Genomic DNA was extracted from plant leaves expressing GA733-Fc (A) and GA733-FcK (B) according to days post-infection (dpi) and leaf position (top, middle, and base leaves), and the targeted genes were amplified and separated on a 1% agarose gel by electrophoresis. The 8–10 dpi samples of base leaves for both GA733-Fc and GA733-FcK were insufficient for PCR. +: positive control (GA733-Fc and GA733-FcK plasmid DNA extracted from Escherichia coli), −: negative control (non-infiltrated plant): 1–10 days post-infiltration samples.

Western blot analysis to determine the expression levels of GA733-Fc and GA733-FcK according to leaf position

Western blot analysis was performed to investigate the differences in expression levels of GA733-Fc and GA733-FcK depending on the leaf position (top, middle, and base leaves) and dpi (Fig. 4). To compare the protein expression level, 2 μl of each sample extracted from equal biomasses was loaded in each western blot lane. In GA733-Fc, there were a 65 kDa and >240 kDa protein bands for the top leaves at 3, 5, and 7 dpi. At 5 dpi, the 65 kDa band density was the strongest among the others in the leaves agroinfiltrated with Ag/pEAQ-GA733-Fc (Fig. 4A). In the middle and base leaves, the 65 kDa band was not detected. However, in the middle leaves at 3–10 dpi, several bands larger than 240 kDa were detected. In plants agroinfiltrated with Ag/pEAQ-GA733-FcK, there were 65 and ~240 kDa protein bands for the top leaves at all dpi (Fig. 4A). The middle leaves also showed the ~65 and ~240 kDa protein bands at 5, 7, 8, and 9 dpi, whereas at 3 and 10 dpi, the bands were barely detected (Fig. 4A right middle). In the base leaves, only the ~240 kDa protein band was detected at 8 dpi. In general, protein expression was higher for GA733-FcK than for GA733-Fc (Fig. 4B). The top leaf position showed relatively high expression regardless of the vector type (Fig. 4B). There was almost no protein expression in the base leaf position (Fig. 4). From the highest to the lowest, the relative protein density levels were ranked top, middle, and base. In general, the highest band density was observed at 5 dpi in the top leaves (Fig. 4).

Figure 4 Comparison of expression levels at various times (1, 3, 5, 7, 8, 9, and 10 days post-infiltration (dpi)) and leaf position (top, middle, and base) by western blot analysis (A), and relative band intensity of GA733-Fc and GA733-FcK protein bands (~65 kDa) with respect to dpi (B).

Western blot analysis was performed to identify GA733-Fc and GA733-FcK protein expression levels in accordance with various dpi using agroinfiltrated leaves of N. benthamiana. GA733-Fc and GA733-FcK were detected using murine anti-GA733 IgGs and anti-murine IgG IgGs conjugated to horseradish peroxidase. +, positive control (mammalian-derived GA733-Fc (GA733-FcM), 70 ng); −, negative control (non-infiltrated plant); 1–10 dpi, samples. (B) The graph shows relative band densities of GA733-Fc (W/O) and GA733-FcK (W) proteins. The crude leaf extract was applied for western blot.

Purification of GA733-Fc and GA733-FcK from agroinfected N. benthamiana plant leaves

GA733-Fc and GA733-FcK transiently expressed in N. benthamiana leaves were purified (Fig. 5). The SDS-PAGE analysis showed that the GA733-Fc and GA733-FcK proteins were successfully purified using protein A affinity chromatography (Fig. 5). Monomer protein bands of the expected size (~65 kDa) were detected for both plant-derived GA733-Fc (GA733-FcP) and GA733-FcKP. The dimeric protein band (~140 kDa) was also observed in the purified GA733-FcKP. The protein band intensity was stronger for GA733-FcK than for GA733-Fc (Fig. 5). Indeed, 1.12 mg of GA733-Fc and 11.82 mg of GA733-FcK were purified from 100 g of infiltrated N. benthamiana leaves (5 dpi).

Figure 5 SDS-PAGE of purified GA733-FcP (W/O) and GA733-FcKP (W) from infiltrated leaves of N. benthamiana plants.

The purified GA733-FcP and GA733-FcKP were analyzed by 10% SDS-PAGE. +, GA733-FcM (1 μg) as a positive control.

ELISA to confirm binding of GA733-FcP and GA733-FcKP proteins to anti-GA733 antibody

ELISA was conducted to compare the binding of GA733-FcP and GA733-FcKP to an anti-GA733 antibody. The 96-well plate was coated with anti-EpCAM antibody (anti-GA733) (50 ng/well), and the purified GA733-FcP and GA733-FcKP proteins were applied (Fig. 6B). Anti-human IgG Fc antibody conjugated to horseradish peroxidase (HRP) was used as a secondary antibody to bind the Fc region of both GA733-FcP and GA733-FcKP (Fig. 6B). The signal from GA733-FcP and GA733-FcKP was higher than that from EpCAM-FcM. The GA733-FcKP showed a higher absorbance compared to the GA733-FcP (Fig. 6).

Figure 6 SDS-PAGE to confirm equal amounts of purified GA733-FcP and GA733-FcKP proteins and sandwich ELISA to assess binding activity of anti-GA733 antibody to both GA733-FcP and GA733-FcKP.

Loading of the same quantity of GA733-FcP and GA733-FcKP proteins in ELISA was confirmed by SDS-PAGE (A). Schematic representation of the sandwich ELISA (B). Results of ELISA for GA733-FcM, GA733-FcP, and GA733-FcKP (C).

Discussion

We demonstrated that the leaf tissue position and days post-infiltration (dpi) affected the transient expression level of recombinant GA733 protein fused to an human immunoglobulin Fc fragment (GA733-Fc) and GA733-Fc fused to an ER retention motif (GA733-FcK) in N. benthamiana plant. For transgenic plant expression, a transformation strategy is required to stably integrate the gene of interest into the plant genome. This transgenic approach has some advantages in terms of scaling up production since the transgene is stably inherited through generations and transgenic seeds can be stored for large-scale cultivation (Chen et al., 2013; Daniell et al., 2009; Peyret & Lomonossoff, 2013). However, it has some disadvantages; for instance, generating plant lines requires a long time with low yields of the target protein. On the contrary, plant transient expression system can overcome these drawbacks of time and yield. For transient expression, plant viral or non-viral vectors are utilized to express genes of interest in plants (Chen et al., 2013). Among these strategies, deconstructed viruses that lack the viral coat protein gene allow for production of heterologous proteins in a few days (Alamillo et al., 2006; Chen et al., 2013; Peyret & Lomonossoff, 2013). Elimination of the region encoding the coat protein enhances the transgene capacity of the virus, enabling it to efficiently produce large recombinant proteins (Chen et al., 2013). In addition, the deconstructed virus vector can replicate in plant cells to a higher level than the full-virus system, allowing a larger amount of recombinant protein to be harvested (Chen et al., 2013).

In the present study, we hypothesized that the transient protein expression levels of GA733-Fc and GA733-FcK varied depending on the leaf position (top, middle, and base) and transfection periods. Thus, the ultimate goal of this study was to optimize the leaf position for agroinfiltration and leaf harvesting time to enhance production of GA733-Fc and GA733-FcK using the plant transient expression system. First, the morphological changes in the leaves (top, middle, and base) at the dpi were investigated. According to the data, 5–7 dpi was the best time for leaf harvesting and purification of recombinant protein since the leaves during these days showed the least damage (i.e., chlorosis, necrosis, and wilting), and the area of the leaves did not reduce when compared to those on the other later days. The loss of chloroplasts results in loss of energy-generating capacity and reduced protein production capacity (Munné-Bosch & Alegre, 2002). Thus, the chlorosis, necrosis, and wilting of leaves induced by Agrobacterium infiltration might affect production of the recombinant protein.

To confirm GA733-Fc and GA733-FcK gene insertion, PCR analysis was conducted. The expected sizes of the amplified PCR bands were observed (GA733-Fc: 1,461 bp, GA733-FcK: 1,473 bp). However, the base leaves of both GA733-Fc and GA733-FcK plants at 8, 9, and 10 dpi were not included in the analysis since the sample mass was insufficient for PCR analysis owing to chlorosis, necrosis, and wilting. All the tested leaf samples produced the expected PCR products, indicating that the infiltration properly delivered both the GA733-Fc and GA733-FcK gene expression cassettes. However, the presence of the transgene in the leaf does not necessarily mean expression of the GA733-Fc and GA733-FcK proteins. Thus, western blotting was carried out to confirm their protein expression. The band of mammalian-derived GA733-Fc was slightly different from plant-derived GA733-Fc and GA733-FcK. It is speculated that the discrepancy is due to difference between mammalian and plant glycosylation, which affects molecular weight, altering protein migration on a western blot (Unal et al., 2008). Accordingly, the protein expression of GA733-Fc and GA733-FcK increased until 5 dpi in the top and middle leaves and then decreased, while the protein expression level in the base leaves was relatively low. In the leaves infiltrated with Agrobacterium carrying the GA733-Fc expression vector, the GA733-Fc protein (~65 kDa) was detected in the upper leaves only at 3, 4, and 5 dpi. The fluctuation of protein expression of GA733-FcK is due to the extreme transient expression of foreign proteins localized in ER during 5~7 dpi resulting in temporary plant cell stress such as ER stress (Howell, 2013) or temporary protease expression and activation (Mandal et al., 2016). In addition, the western blot analysis showed that the protein expression level was higher in the leaves infiltrated with Agrobacterium carrying the GA733-FcK expression vector than in the leaves infiltrated with the GA733-Fc expression vector. These results indicate that the KDEL ER retention signal affected the production level of the GA733-Fc fusion protein in the transient plant expression system. In previous studies, KDEL tagging to recombinant antibodies and antigens localized and accumulated them in the ER, eventually increasing their production levels in transgenic plants (Kang et al., 2016; Lee et al., 2013, 2020a; Park et al., 2020; So et al., 2013; Song et al., 2018). Indeed, N-glycosylation analysis showed that GA733-FcK had mainly oligo-mannose compared to GA733-Fc, indicating the KDEL induced ER retention of GA733-FcK (Fig. S1).

Both GA733-Fc and GA733-FcK expression levels in the leaves decreased in the order of top, middle, and base. The bottom leaves become the main leaves first after germination. These leaves are older with fewer chloroplasts, resulting in lower energy and protein production than the middle and top leaves, which are younger (Munné-Bosch & Alegre, 2002). Such low recombinant protein expression is probably related to the low concentration of total soluble protein in the old leaves (Halfhill et al., 2003).

For purification of both GA733-Fc and GA733-FcK, protein A affinity chromatography was utilized since the Fc domain was fused to the GA733 protein (Lee et al., 2020a; Park et al., 2015, 2020; Shin et al., 2019). SDS-PAGE showed that the purified GA733-FcK had stronger band intensity than the purified GA733-Fc. The concentration of the purified GA733-Fc and GA733-FcK proteins was 0.112 and 1.82 mg/ml, respectively, representing 16.25 times higher production of GA733-FcK than GA733-Fc. The enhanced protein production of GA733-FcK in the current study is consistent with that in a previous study, where the transient expression of ER-targeted human interleukin-6 represented 7% of the total soluble protein (TSP) in N. benthamiana (Munné-Bosch & Alegre, 2002). We speculate that the ER-retained GA733-Fc protein molecules interacted with ER-resident chaperones to promote correct folding (Benchabane et al., 2008) and were less exposed to proteases in the ER (Vitale & Pedrazzini, 2005) in transient plant system.

In this present study, we confirmed that the protein expression varied with the dpi. After agroinfiltration of the GA733-FcK vector, the top leaves began to express the protein at 3 dpi, with strong expression from 5 to 9 dpi, except for 7 dpi. For the middle leaves, the highest expression was observed at 5 dpi. These results suggest that the best leaf harvesting time for purification of GA733-FcK proteins is 5 dpi and that GA733-Fc can be transiently expressed using the CPMV virus-based vector with an optimized transfection period and leaf position. According to the current results, this harvesting time is applicable for top leaves infiltrated with Agrobacterium carrying the GA733-Fc vector. Previous studies also showed high expression of recombinant proteins at 4–6 dpi (Norkunas et al., 2018). The leaf tissue position and dpi affect both GA733-Fc and GA733-FcK recombinant protein expression. Thus, to optimize protein expression, the top leaves should be infiltrated and then harvested from 5 to 9 dpi, and not before 3 dpi and after 9 dpi.

Sandwich ELISA was performed to detect the binding of each purified GA733-Fc and GA733-FcK protein to an anti-GA733 antibody. The resulting absorbance of GA733-FcK was stronger than that of GA733-Fc. These results suggest that the antibody is able to bind GA733-FcK better than GA733-Fc, and they are similar to our previous studies where recombinant proteins tagged with KDEL showed better interactions with antibodies regardless of the specific antigen or antibody (Kang et al., 2016; Lee et al., 2013; So et al., 2013; Song et al., 2018). We speculate that ER-retained recombinant proteins are more likely to be correctly folded or favorably glycosylated for better binding activity than cytosolic or apoplastic proteins. In the surface plasmon resonance (SPR) analysis (Fig. S2), in addition, GA733-FcK had higher affinity for FcγRⅠ (CD64) than GA733-Fc, indicating that the Fc domain of GA733-FcK interacts better with FcγRⅠ (CD64) than that of GA733-Fc.

Conclusions

In this study, we determined the optimal leaf position and dpi for the transient expression of GA733-Fc and GA733-FcK. In addition, we confirmed that KDEL tagging of GA733-Fc enhanced its production in the transient plant expression system. Taken together, the Fc fusion protein GA733-Fc can be expressed using a CPMV virus-based vector in transient plant system. Plants are in the spotlight as an excellent therapeutic protein production platform. We have expressed and purified colorectal cancer vaccine recombinant protein GA733-Fc and GA733-Fc with KDEL (GA733-FcK) by transient expression system using pEAQ-HT vector with reduced labor than transgenic expression system. By using our currently described optimized GA733-Fc and GA733-FcK transient expression condition, it is expected that mass production of GA733-Fc and GA733-FcK would be realized in the future.

Supplemental Information

Supplemental Information 1 Surface plasmon resonance(SPR) analysis to confirm the binding affinity of GA733-Fc P (a) andGA733-FcK P (b) to FcγR Ⅰ (CD64).

Click here for additional data file.

Supplemental Information 2 PCR data of the base leaf infiltrated with Agrobacterial inoculum carryingpEAQ-GA733-Fc (Ag/pEAQ-GA733-Fc) showing the existence of GA733-Fc in plant topleaf.

The genomic DNA fragments were extracted from infiltrated leaves located base position, amplified and separated on a 1% agarose gel using electrophoresis. Positive control (+), GA733-Fc plasmid DNA extracted from Escherichia coli negative control (-), genomic DNA extracted from non-infiltrated Nicotiana benthamiana plant top leaves; 1-10 dpi, genomic DNA extracted from infiltrated Agrobacterial inoculum carrying pEAQ-GA733-Fc (Ag/pEAQ-GA733-Fc).

Click here for additional data file.

Supplemental Information 3 PCR data of the base leaf infiltrated with Agrobacterial inoculum carrying pEAQ-GA733-Fc (Ag/pEAQ-GA733-Fc) showing the existence ofGA733-Fc in plantmid leaf.

The genomic DNA fragments were extracted from infiltrated leaves located base position, amplified and separated on a 1% agarose gel using electrophoresis. Positive control (+), GA733-Fc plasmid DNA extracted from Escherichia coli; negative control (-), genomic DNA extracted from non-infiltrated Nicotiana benthamiana plant mid leaves; 1-10 dpi, genomic DNA extracted from infiltrated Agrobacterial inoculum carrying pEAQ-GA733-Fc (Ag/pEAQ-GA733-Fc).

Click here for additional data file.

Supplemental Information 4 PCR data of the base leaf infiltrated with Agrobacterial inoculum carrying pEAQ-GA733-Fc (Ag/pEAQ-GA733-Fc) showing the existence ofGA733-Fc in plant base leaf.

The genomic DNA fragments were extracted from infiltrated leaves located base position, amplified and separated on a 1% agarose gel using electrophoresis. Positive control (+), GA733-Fc plasmid DNA extracted from Escherichia coli; negative control (-), genomic DNA extracted from non-infiltrated Nicotiana benthamiana plant base leaves; 1-10 dpi, genomic DNA extracted from infiltrated Agrobacterial inoculum carrying pEAQ-GA733-Fc (Ag/pEAQ-GA733-Fc).

Click here for additional data file.

Supplemental Information 5 PCR data of the base leaf infiltrated with Agrobacterial inoculum carrying pEAQ-GA733-FcK (Ag/pEAQ-GA733-FcK) showing the existence of GA733-FcK in plant topleaf.

The genomic DNA fragments were extracted from infiltrated leaves located base position, amplified and separated on a 1% agarose gel using electrophoresis. Positive control (+), GA733-FcK plasmid DNA extracted from Escherichia coli; negative control (-), genomic DNA extracted from non-infiltrated Nicotiana benthamiana plant top leave; 1-10 dpi, genomic DNA extracted from infiltrated Agrobacterial inoculum carrying pEAQ-GA733-FcK (Ag/pEAQ-GA733-FcK).

Click here for additional data file.

Supplemental Information 6 PCR data of the base leaf infiltrated with Agrobacterial inoculum carrying pEAQ-GA733-FcK (Ag/pEAQ-GA733-FcK) showing the existence of GA733-FcK in plant Midleaf.

The genomic DNA fragments were extracted from infiltrated leaves located base position, amplified and separated on a 1% agarose gel using electrophoresis. Positive control (+), GA733-FcK plasmid DNA extracted from Escherichia coli; negative control (-), genomic DNA extracted from non-infiltrated Nicotiana benthamiana plant base leaves; 1-10 dpi, genomic DNA extracted from infiltrated Agrobacterial inoculum carrying pEAQ-GA733-FcK (Ag/pEAQ-GA733-FcK).

Click here for additional data file.

Supplemental Information 7 PCR data of the base leaf infiltrated with Agrobacterial inoculum carrying pEAQ-GA733-FcK (Ag/pEAQ-GA733-FcK) showing the existence of GA733-FcK in plant base leaf.

The genomic DNA fragments were extracted from infiltrated leaves located base position, amplified and separated on a 1% agarose gel using electrophoresis. Positive control (+), GA733-FcK plasmid DNA extracted from Escherichia coli; negative control (-), genomic DNA extracted from non-infiltrated Nicotiana benthamiana plant mid leaves; 1-10 dpi, genomic DNA extracted from infiltrated Agrobacterial inoculum carrying pEAQ-GA733-FcK (Ag/pEAQ-GA733-FcK).

Click here for additional data file.

Supplemental Information 8 Western blot of the base leaf infiltrated with Agrobacterial inoculum carrying pEAQ-GA733-Fc (Ag/pEAQ-GA733-Fc) showing the existence ofGA733-Fc in plant leaf. The bottom blot was used for the Figure 3A top (left panel).

Lane 1, protein marker; Lane 2, positive control (+), mammalian-derived GA733-Fc (EpCAM-Fc M), 70 ng; -, negative control (non-infiltrated Nicotiana benthamiana plant base leaves); 1-10 dpi, samples.

Click here for additional data file.

Supplemental Information 9 Western blot of the base leaf infiltrated with Agrobacterial inoculum carrying pEAQ-GA733-Fc (Ag/pEAQ-GA733-Fc) showing the existence ofGA733-Fc in plant leaf. The bottom blot was used for the Figure 3A Mid (left panel).

Lane 1, protein marker; Lane 2, positive control (+), mammalian-derived GA733-Fc (EpCAM-Fc M), 70 ng; -, negative control (non-infiltrated Nicotiana benthamiana plant base leaves); 1-10 dpi, samples.

Click here for additional data file.

Supplemental Information 10 Western blot of the base leaf infiltrated with Agrobacterial inoculum carrying pEAQ-GA733-Fc (Ag/pEAQ-GA733-Fc) showing the existence ofGA733-Fc in plant leaf. The bottom blot was used for the Figure 3A Base (left panel).

Lane 1, protein marker; Lane 2, positive control (+), mammalian-derived GA733-Fc (EpCAM-Fc M), 70 ng; -, negative control (non-infiltrated Nicotiana benthamiana plant base leaves); 1-10 dpi, samples.

Click here for additional data file.

Supplemental Information 11 Western blot of the base leaf infiltrated with Agrobacterial inoculum carrying pEAQ-GA733-FcK (Ag/pEAQ-GA733-FcK) showing the existence of GA733-Fc in plant leaf. The bottom blot was used for the Figure 3A Top (right panel).

Lane 1, protein marker; Lane 2, positive control (+), mammalian-derived GA733-Fc (EpCAM-Fc M), 70 ng; -, negative control (non-infiltrated Nicotiana benthamiana plant base leaves); 1-10 dpi, samples.

Click here for additional data file.

Supplemental Information 12 Western blot of the base leaf infiltrated with Agrobacterial inoculum carrying pEAQ-GA733-FcK (Ag/pEAQ-GA733-FcK) showing the existence of GA733-Fc in plant leaf. The bottom blot was used for the Figure 3A Mid (right panel).

Lane 1, protein marker; Lane 2, positive control (+), mammalian-derived GA733-Fc (EpCAM-Fc M), 70 ng; -, negative control (non-infiltrated Nicotiana benthamiana plant base leaves); 1-10 dpi, samples.

Click here for additional data file.

Supplemental Information 13 Western blot of the base leaf infiltrated with Agrobacterial inoculum carrying pEAQ-GA733-FcK (Ag/pEAQ-GA733-FcK) showing the existence of GA733-Fc in plant leaf. The bottom blot was used for the Figure 3A Base (right panel).

Lane 1, protein marker; Lane 2, positive control (+), mammalian-derived GA733-Fc (EpCAM-Fc M), 70 ng; -, negative control (non-infiltrated Nicotiana benthamiana plant base leave); 1-10 dpi, samples.

Click here for additional data file.

Supplemental Information 14 Relative band intensity of GA733-Fc and GA733-FcK protein bands with respect to dpi.

Data for both GA733-Fc and GA733-FcK protein band density obtained from the western blot analyses. The graph was obtained from the data which calculating the average value of triple measurement per each case including standard deviation, showing relative band densities of GA733-Fc (W/O) and GA733-FcK (W) proteins. These were used for Figure 3B. The vertical axis values represent the relative density of western blot results at each location, and lateral axis represent days post infiltration (Dpi).

Click here for additional data file.

Supplemental Information 15 SDS-PAGE analysis of purified GA733-Fc P (W/O)and GA733-FcK P (W) from infiltrated leaves of N. benthamiana plants.

Lane 1, protein marker; Lane 2, positive control (+), mammalian-derived GA733-Fc (EpCAM-Fc M), 100 μg; Lane 3, The purified GA733-Fc P; Lane 4 The purified GA733-FcK P. Commercial protein A resin column (Sepharose 4 Fast Flow, GE Healthcare, Sweden, NJ). The gel was stained with Coomassie blue staining solution [10% acetic acid (v/v), 30% methanol (v/v), and 0.01% Coomassie blue (w/v)] at room temperature for 30 min with shaking. The gel was destained with destaining solution (10% acetic acid and 30% methanol) thrice for 30 min each withshaking.

Click here for additional data file.

Supplemental Information 16 SDS-PAGE gel with GA733-Fc M, GA733-Fc P, and GA733-Fc P to confirm the equal amount for ELISA.

The Coomassie blue stained SDS-PAGE gel with GA733-Fc M, GA733-Fc P, and GA733-Fc P to confirm the equal amount for ELISA. This was used for Figure 5A. Loading of the same quantity of GA733-Fc P and GA733-FcK P proteins in ELISA was confirmed by SDS-PAGE. Lane 1, protein marker; Lane 2, positive control (+), mammalian-derived GA733-Fc (GA733-Fc M); Lane 3, GA733-Fc P; Lane 4, GA733-FcK P.

Click here for additional data file.

Supplemental Information 17 ELISA to confirm the ability of GA733-Fc and GA733-FcK to bind anti-GA733 antibody.

The ELISA results with GA733-Fc M, GA733-Fc P, and GA733-Fc P were used for Figure 5C. The experiment was repeated a total of 6 times. The graph result was obtained by calculating the average value including standard deviation. The vertical axis values (450 nm) represent the absorvance of each case, and lateral axis represent positive control (GA733-Fc M), purified GA733-Fc P, and GA733-FcK P.

Click here for additional data file.

We thank Professor George Lomonossoff (John Innes Centre, Norwich, United Kingdom) for providing the plant viral pEAQ-HT vector. This study was conducted in Department of Medicine, Therapeutic Protein Engineering Lab, College of Medicine, Chung-Ang University.

Additional Information and Declarations

Competing Interests

Author Contributions

Data Availability

The authors declare that they have no competing interests.

Kibum Kim conceived and designed the experiments, performed the experiments, analyzed the data, authored or reviewed drafts of the paper, and approved the final draft.

Yang Joo Kang performed the experiments, analyzed the data, prepared figures and/or tables, authored or reviewed drafts of the paper, and approved the final draft.

Se Ra Park performed the experiments, analyzed the data, authored or reviewed drafts of the paper, and approved the final draft.

Do-Sun Kim conceived and designed the experiments, prepared figures and/or tables, and approved the final draft.

Seung-Won Lee performed the experiments, analyzed the data, authored or reviewed drafts of the paper, and approved the final draft.

Kinarm Ko conceived and designed the experiments, authored or reviewed drafts of the paper, and approved the final draft.

Daniel Ponndorf conceived and designed the experiments, prepared figures and/or tables, and approved the final draft.

Kisung Ko conceived and designed the experiments, authored or reviewed drafts of the paper, and approved the final draft.

The following information was supplied regarding data availability:

Raw data are available in the Supplemental Files.

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
