# Peer review of "Effect of leaf position and days post-infiltration on transient expression of colorectal cancer vaccine candidate proteins GA733-Fc and GA733-FcK in Nicotiana benthamiana plant"

_PeerJ, doi:10.7717/peerj.10851_

## Round 0.1 · original submission · Minor Revisions

Please respond to the reviewer comments with a point by point response rebuttal. We look forward to receiving your revised manuscript.

Reviewer 1 ·

Basic reporting

This manuscript is well written and the introduction is comprehensive. The materials & methods section needs a minor update but results and discussion summarizes the body of work clearly. Details provided in the figures are quite helpful.

Experimental design

In general, the experimental design was good but a minor clarification is needed.
A. Line227-228: The statement does not clarify the procedure for extraction. Is the sample used for western-blot analysis a crude extract or purified preparation? Please provide the details.

Validity of the findings

1) Line 229-231: Please clarify: Does the author is intending to say in the context of 70 kDa band only?
2) Figure 4. The differences observed in the protein band position between + (mammalian expressed GA733-Fc) and Nicotiana benthamiana expressed proteins is due to differences in the glycosylation due to different expression platforms?
3) Can an explanation be provided for the differences observed in the molecular weight of the + control and the test samples? In the western-blot (Figure 3), a band corresponding to 70 kDa is seen whereas in Figure 4 and Figure 5 A, the band lies in between 50 – 70 kDa for + control as well as the test samples. What is the reason for this shift observed between the figures?
4) Line355-356: In my opinion, the study does not support the claim that KDL tagging of GA733-Fc improved its biological activity. This claim (biological activity) needs further support by performing cell or animal-based studies.
5) Line359: I would recommend removing the word “efficiently” from the conclusions as the productivity of the protein has not been reported in the manuscript.
6) It will be beneficial for the reader to and promote this expression system, the author could compare productivity (total product/gm of leaves) of this protein achieved in this study with other expression systems (mammalian).

·

Basic reporting

The manuscript was written clearly and is easily understandable. The goal of this study was mentioned with supporting background information. The introduction was concise leading directly to the gap in the knowledge and the goal of the study is to fill that gap. The article was written strictly according to guidelines from standard sections. My only suggestion for the results section is to increase the resolution of figure 4. The raw data from the manuscript is very useful to me in assessing the quality of results from well-defined experiments. All results support the main point of this manuscript which is to identify the spatial and temporal expression of tumor-associated antigen GA733 in a plant-based transient expression system.

Experimental design

The question that was answered in this manuscript is relevant to the scope of PeerJ. Recombinant protein production using a plant-based system is well-known. However, generating a transgenic plant with stably expressing recombinant protein is a time-consuming process with unpredictable results. Therefore, this manuscript addresses these issues using a transient plant-based expression system. Besides, the authors screened the spatial and temporal variables for maximum expression of tumor-associated antigens GA733-Fc and GA733-Fck in a plant-based transient expression system. Overall, this will be very useful in bypassing the mammalian cell expression system and significantly reduces the recombinant protein production expenses.
The methods were described in great detail and also referenced to obtain the materials required for performing reproducibility studies. However, additional details would have assisted in the context comparing the expression of GA733-Fc and GA733-FcK:
1) The manuscript concluded that GA733-FcK was expressed in higher amounts when compared to GA733-Fc. However, the exact amount of agrobacteria injected into bethamainan leaf was not provided. The small differences in the number of agrobacteria injected between GA733-Fc and GA733-FcK could lead to a difference in the protein yields as mentioned in this manuscript.
2) The authors compared expression from leaves at different positions on the plant and concluded that top leaves produced more protein than base leaves. However, the study did not clearly state if the agroinfiltration had been performed on the same age of the leaf from different positions of the plant? Based on the text from results and discussion sections, it looks like the authors didn’t consider the age of leaf at a particular position (Top, Middle, and Base) before agroinfiltration and protein expression analysis.

Validity of the findings

Previous publications explored the plant-based expression of GA733. Also, these studies provided evidence that the recombinant GA733 expressed in the plant-based system was functional and economical to produce. However, the novelty of this study arises from two parts. First, use of a transient system instead of a laborious stable expression system and second, establishing the spatial and temporal conditions for maximum yield of recombinant GA733.
The presence of GA733-Fc and GA-733-FcK genes in benthamiana leaves was analyzed with PCR and protein expression was shown using SDS-PAGE. The specificity of expressed proteins was analyzed using the western blot technique. A critical step in quality control of a recombinant protein is its functional assessment. This manuscript provided ELISA data supporting 50 % of the answer to its functional assessment. In my opinion, an in vivo functional assessment of recombinant expressed GA733-Fc and GA733-FcK proteins would have been a significant boost to the impact of this manuscript. The data provided in this manuscript is robust and proper controls had been used in all the experiments. For example, the authors assessed data from 4 leaves for calculating relative density results in figure 3B. However, a statistical analysis would have been useful in comparing the relative temporal expression of proteins. Also, supporting data comparing the spatial expression of proteins could have been generated by analyzing the absolute amount of protein produced from the same quantity of starting material. For example, the absolute total amount of proteins can be obtained by performing size-exclusion chromatography on Protein A elution material. The authors made conclusions based on results from well-designed experiments. Speculation had been made in the discussion section to answer the lower expression from base leaves. No supporting data is shown to compare either chlorosis or necrosis between leaf positions (top vs middle vs base).

Additional comments

1) In figure 3B, it was shown that the expression of GA733-FcK was decreased from day 5 to day 7 and increased again. Did you get a chance to examine the reason for these changes?
2) What are the 240 and 30 kDa bands that showed up in SDS-page and western blot analysis?

·

Basic reporting

The article uses clear, unambiguous, and professional English.
Literature references are sufficient with field background/context provided.
Article structure, figures, tables, etc. are professional.
Article is self-contained with relevant results to hypotheses.

Experimental design

Original research within the scope and aim of the journal.
Research question well defined and meaningful. Knowledge gaps explained and filled.

Investigation could have been more rigorous or at least the methods described more clearly in some cases. See the below comments for details.

Line 68-70: According to Peyret & Lomonossoff 2013 the pEAQ-HT vector allows for protein production at a maximum reported amount of ~0.5g/kg. However, Diamos et al 2020 reports production of monoclonal antibodies at “3-5 g/kg leaf fresh weight or ~50% total soluble protein.” Using the geminiviral expression system. This holds true with both VLPs and Toxoid Fusions produced using the same expression system. While it is perhaps not as easily utilized, this data shows the below statement to be of issue

“This deconstructed vector lacks the viral coat protein and has been shown to allow higher and faster recombinant protein production than other plant expression systems.”

Line 113-115: It is customary to report final OD600 of the infiltrated solution as was done in Sainsbury et al. 2009. While an OD range of 0.3-0.4 was measured before cells were spun down and resuspended inaccuracies during this step could lead to inaccuracies in the final amount of protein expressed.

Line 328-330: “except for 7 dpi” Protein production does not generally fall off during the expression period and then increase again according to all data I’ve gathered when doing time courses. It was my assumption that all samples were taken in triplicate, however this is not clearly stated in Figure 3 or in the text. If this was not done, a major short coming of this study could be that there is a large difference in protein expression among leaves/plants even if samples are taken from the same area (top, middle, or base). While the general trend is evident and supported, if the samples were not taken in triplicate, or were not taken from the same leaf, on the same plant, for each DPI, then I highly suggest noting these shortcomings in the paper or repeating this part of the experiment. While repeating the experiment would take a lot of time, and the benefit would be minimal (due to the general trend) it is important that this question of replication in the samples is addressed.

I suggest minor revisions be made to fix this issue.

Validity of the findings

The findings from this article are novel and well-articulated both through the text and figures.

Most data have been provided and is robust in general. See the below notes.

Line 328-330 as stated above.

Line 308-310: Is there a good reason why this data is not shown? If so, please state why (such as being part of a data set to be published in the future).

Conclusions are well stated, connected to the hypothesis, and limited to supporting results.

All speculation is based on the experimental evidence at hand and is clearly stated as such.

Additional comments

Line 32: The statement of difference in expression between GA733-FcK and GA73-FcK would benefit from the mg/g (or mg/ml) leaf tissue being reported, or a percentage.

Line 238-241: The experimentation used to determine the protein production in each major leaf group (top, middle, base) is quite robust from a design point of view (even if there is the issue of multiple samples being tested) and confirms what has been a general consensus among several plant-based protein expression groups using N. Benthamiana known to myself.

The paper is very clear throughout with the only errors being what I’ve noted herein. It is a very good paper and was worthwhile to delve into.

---

## Round 0.2 · accepted · Accept

Thanks for addressing the changes suggested by the reviewers.

Reviewer 1 ·

Basic reporting

I thank Authors for addressing reviewer's comments and the manuscript is well written and the introduction is comprehensive.

Experimental design

Once again, I thank Author's considering my comments and addressing the issues.

Validity of the findings

Author's did addressed my concerns with the claims and happy with their response.

·

Basic reporting

No comment

Experimental design

No comment

Validity of the findings

No comment

Additional comments

Thank you for taking the time to correct all the issues I had with the manuscript. It is a fine paper.